# Comparative Analysis of Physiological and Biochemical Responses to Salt Stress Reveals Important Mechanisms of Salt Tolerance in Wheat

**DOI:** 10.3390/ijms26083742

**Published:** 2025-04-16

**Authors:** Tahar Taybi, Norah Alyahya

**Affiliations:** 1School of Natural and Environmental Sciences, Newcastle University, Newcastle upon Tyne NE1 7RU, UK; nmushbbab@kku.edu.sa; 2Department of Biology, Faculty of Science, King Khalid University, Abha 62521, Saudi Arabia

**Keywords:** wheat, Qiadh, Najran, salt stress, growth, yield, proline, sugars, phenolics, transcript levels

## Abstract

Salinity tolerance varies among wheat (*Triticum aestivum*) tissues and cultivars. This study investigated the impact of salt stress on two Saudi wheat cultivars, Qiadh and Najran. Growth parameters (fresh weight, dry weight and plant length), biochemical responses (proline, soluble sugars, starch and organic acids contents) and antioxidants (phenolics content), as well as gene responses, were assessed in the control and salt (NaCl)-treated plants. A distinctive variation was observed between the two cultivars. Najran was the most tolerant to salt stress. Salt stress caused a dramatic decline in growth parameters in both cultivars; however, Qiadh exhibited the highest reduction in growth and yield. Differential increase in metabolite content occurred in the two cultivars under salinity stress, with levels varying between cultivars and roots and shoots. Transcripts for genes involved in the production of proline, sugars, starch and phenolics increased in shoots and roots, to higher levels in Qiadh compared to Najran. Interestingly, transcript levels for genes involved in sugar and starch synthesis declined in Najran roots. The obtained results confirm that different wheat cultivars employ varying mechanisms to alleviate the harmful effects of salt stress. The salt-tolerant Najran cultivar might constitute a genetic source for breeding enhanced salt tolerance in other wheat cultivars.

## 1. Introduction

In recent years, climate change has had a direct impact on agricultural production and quality of yields by increasing the frequency and severity of several environmental stresses [1]. Salt stress is one of these stresses impacting 20% of the world’s cultivable land and contributing to an approximate 50% decrease in crop outputs [2,3]. Wheat is the second-most globally cultivated crop and is a main source of vegetable proteins and daily calories required for human consumption [4]. Wheat in Saudi Arabia has a major role in the baking industry and its production was around 3–4 million tonnes in the beginning of 1990s. However, wheat yield has decreased to 2.63 million tonnes since 1993 due to various limiting environmental factors including salt stress [5]. Soil salinity is becoming more severe in the brackish water-irrigated lands constituting a global threat for food production. High salinity represents a considerable constraint to crop production limiting the yield and quality of the crop [6]. Therefore, it is becoming a hard challenge to boost crop output and meet food security under increasing salinity conditions.

High levels of sodium chloride in soil interfere with plant growth, imposing various types of stresses, such as osmotic and ionic stresses. Plants have evolved several physiological and biochemical mechanisms as essential responses to these stresses. Stomatal closure has been reported for being one of the most common responses to osmotic stress. The stomatal closure results in a reduction in plant biomass as a consequence of carbon starvation, and over a period of time, can lead to early senescence of mature leaves, which might be followed by plant death [7]. It has been revealed that accelerated senescence is an adaptive method by which stressed plants reduce their canopy size and consume carbon and nutrients in their reproductive parts to produce seeds. Although this response is efficient for next generation survival, it leads to a yield decline in annual crops [8]. In addition, osmotic adjustment within stressed plant cells has been evidenced as a crucial contributor mechanism in acclimation to salt stress in various plant species; sugar beet [9], cotton [10], durum wheat [11] and bean [12]. Under salt stress, plants osmotically adjust to maintain cellular turgor and the structural integrity of membranes [13]. Moreover, antioxidant defense systems are another important protective mechanism that is induced under salt stress, which prevents the cellular damage caused by salt-induced ROS accumulation [14].

Wheat has been widely considered to be a moderately salt-tolerant plant and its tolerance and responses to salinity stress vary among different tissues and cultivars [15]. Salt tolerance is among the most physiologically complex traits in plants; it is controlled by a number of mechanisms, some of which are specific to salt stress and some of which are common to other stress types. It seems that plants vary in the set of mechanisms operating under salt stress, depending on the species and cultivar [16]. Wheat cultivars, which have been developed through selective breeding programs and genetic selection, demonstrate diverse levels of tolerance to environmental stresses, including salt stress, indicating great genetic diversity within the species [17]. The exploration of variations in salt tolerance among different wheat cultivars has become a crucial objective in modern agricultural research [15] as a strategic solution to enhance wheat production in salt-affected areas. Various cultivars of wheat have been documented to exhibit differences in their growth and yield outcomes under salinity, showing different levels of tolerance to salt stress [18,19]. Some wheat cultivars show remarkable resilience, exhibiting a minimal decrease in growth and yield upon exposure to salinity conditions. In contrast, other cultivars are more susceptible to stress and thus suffer significant losses. Reference. [20] investigated the variations in physiological and biochemical responses between two wheat cultivars under salt stress. They found that the salt-tolerant Suntop cultivar exhibited lower reductions in growth and photosynthetic efficiency and higher activities of antioxidant enzymes, exhibiting higher tolerance to salinity compared to the salt-sensitive cultivar (Sunmate).

Hundreds of wheat cultivars have been grown in different regions of Saudi Arabia for hundreds of years. Many of these cultivars have evolved adaptations to prevailing local conditions; thus, they represent an invaluable germplasm resource that requires proper characterization. Few studies have been conducted to evaluate the tolerance responses of different Saudi wheat cultivars to abiotic and biotic stresses. For examples, responses of agronomic performance and yield potentials to water stress [21,22,23], growth and physiological responses to heat stress [24], responses to pathogen attacks [25] and responses of morphological traits to gamma irradiations [26]. Very limited studies have attempted to investigate the different responses of typical Saudi wheat to salt stress [27,28,29]. Therefore, this investigation was conducted to characterize the differential responses to salt stress in two Saudi wheat cultivars, Najran and Qiadh, in cultivation in different regions of the Kingdom to potentially reveal the underlying mechanisms for salt tolerance in wheat. The study investigated variation in the physiological and biochemical responses, as well as antioxidant scavenging capacity via phenolics accumulation among the two cultivars. The obtained knowledge constitutes an important addition towards understanding the different salt-tolerance mechanisms in wheat and potentially help to develop wheat cultivars with higher salt tolerance.

## 2. Results

### 2.1. Plant Growth and Development

Growth performance of the two *T. aestivum* cultivars under salt stress and control conditions was evaluated by measuring different parameters including RFW, SFW, RDW, SDW, RL and SL. All these growth parameters were at similar levels in the examined cultivars under unstressed conditions; however, significant differences appeared under salt stress between cultivars (Figure 1). Fresh and dry weight under salinity treatment in both roots and shoots was significantly lower than those in the control plants. The two cultivars have shown relatively similar reductions in both Fresh Weight and Dry Weight in roots and shoots under salt stress consisting of 9- to 10-fold reductions, respectively, (Figure 1A,B). In contrast, root length similarly decreased in Qiadh (38%), and Najran (39%), the same for shoot length, which similarly reduced in Qiadh (38%) and Najran (36%) (Figure 1C).

### 2.2. Grain Yield

As shown in Figure 2A,B, there was a significant difference between the two wheat cultivars regarding their seed and spike numbers. Qiadh had the largest number of spikes in the control and NaCl-treated plants (three spikes), and the largest number of seeds in control plants (54 seeds), however it had the smallest number of seeds in NaCl-treated plants (17 seeds). On the other hand, Najran had the lowest number of spikes (one spike) and seeds (17 seeds) in control plants, compared to salt-treated plants. In addition, salt treatment had a positive effect on spike and seed number in Najran, whereas Qiadh displayed a negative salt effect on both parameters. This result reveals that Qiadh was the most affected cultivar by salt stress, as the number of seeds decreased dramatically (*p* < 0.001) and the number of spikes reduced slightly (*p* > 0.05), while the seed number increased slightly, and the number of spikes increased significantly (*p* < 0.01) in the Najran cultivar.

The number of seeds was not the only condition affected under saline conditions, but also the weight of seeds, where both wheat cultivars exhibited a significant decline in seed weight (*p* < 0.001) in comparison to the control plants (Figure 2C). The produced grains of the two wheat cultivars were germinated under the control (0 mM NaCl) and saline (200 mM NaCl) conditions. As depicted in Figure 2D, Qiadh shows a higher reduction (28%) in seed germination rate under salt stress compared to Najran (8%).

### 2.3. Proline Content

Plants subjected to salt treatment displayed an increased accumulation of Proline in roots and shoots compared to control plants. As shown in Figure 3, under un-stressed conditions the two wheat cultivars had little Proline content to be measured in their roots and shoots except Qiadh which had a tiny amount of Proline only in its shoot tissues (0.01 µg mg^−1^ DW). Salt stress induced an important increase in proline content in both the roots and shoots of the two wheats (Figure 3). However, a significant difference in free Proline content in root and shoot tissues was observed between the two wheat cultivars under salt treatment (*p* < 0.01). In response to salt stress, Qiadh had a lower proline content in its roots compared to Najran, at 0.01 and 0.17 µg mg^−1^ DW, respectively. In contrast, Qiadh had the larger proline content in its shoots than Najran, at 0.66 and 0.39 µg mg^−1^ DW, respectively (Figure 3).

### 2.4. Soluble Sugars and Starch Levels

There was no significant difference between levels of soluble sugars in the two wheat cultivars under unstressed conditions (Figure 4A). However, the content of soluble sugars differed significantly among salt-treated plants of the two cultivars. It increased under salt stress in roots by 4.5-fold in Qiadh and 7.6-fold in Najran cultivar, and in shoots, it increased by 4.9-fold in Qiadh and 1.9-fold in Najran cultivar (Figure 4A).

In contrast, there was a significant variation between the two wheat cultivars regarding the starch levels in roots and shoots of control plants (Figure 4B). Qiadh displayed almost no starch in roots, Najran exhibited 0.06 µg mg^−1^ DW of starch in its roots, while 0.04 and 0.16 µg mg^−1^ DW accumulated in the shoots of the two cultivars, respectively. Salt stress led to a significant boost in starch accumulation, with Qiadh having the highest increase in starch levels in both roots and shoots, respectively, relative to the control.

### 2.5. Total Organic Acids

Levels of total organic acids in the roots and shoots of control plants were significantly different between the two wheat cultivars (*p* < 0.01). Salt stress resulted in a big increase in total organic acids in the two wheat cultivars (*p* < 0.05) (Figure 5). In the Najran cultivar, which has shown higher salt tolerance, salt stress increased the content of organic acids 6.3 folds in the root, whereas in Qiadh, which exhibited less stress tolerance, only a 1.7-fold increase was measured under salt stress. Salt stress induced the highest increases in total organic acids of 35 folds and 19.3 folds in the shoots of Qiadh and Najran, respectively.

### 2.6. Phenolics Content

NaCl treatment significantly enhanced the production of phenolics in root and shoot tissues of Najran, and Qiadh wheats. As seen in Figure 6, a pronounced increase in phenolics content was observed in the roots and shoots of salt-treated plants of the two cultivars compared to the control. Higher levels of phenolics content of 3.48 and 3.20 nmol.mg^−1^ DW were recorded in Najran, whereas Qiadh showed lower values of phenolic compounds of 1.83 and 1.87 nmol.mg^−1^ DW in their roots and shoots, respectively.

### 2.7. Gene Expression

To monitor the effect of salt stress on the expression of key genes in the production of the measured metabolites, transcript levels of genes encoding phenylalanine ammonia lyase (PAL), delta1-1-pyrroline-5-carboxylate synthase (P5CS1), sucrose synthase (SUS) and starch synthase (STSR) were measured by semi-quantitative RT-PCR in the shoots and roots of control, Qiadh and Najran wheat plants, watered with tap water and salt-treated plants, watered with 200 mM NaCl. As shown in Figure 7, salt stress resulted in an increase in transcript levels of all monitored genes in both the shoots and roots of Qiadh with the exception of the Sucrose synthase gene, which had only a minor increase in shoots. Similarly, transcript levels for genes encoding SUS and STS increased in shoots of salt stressed Najran plants. In contrast, there was a decrease in transcript levels for the two genes in roots (Figure 7). Transcript levels for PAL and P5CS1 genes remained unchanged in both the shoots and roots of salt-stressed Najran plants (Figure 7).

## 3. Discussion

The variability in salt-stress responses among wheat cultivars might be attributed to differences in the genetic background, these differences control salt-stress perception and signaling pathways, osmotic adjustment capacity, ion transport and compartmentalization and the activation of stress-response genes [30,31].

### 3.1. Plant Growth, Salt Stress Reduced Differentially the Growth of the Two Wheat Cultivars

Salinity alters plant growth and development, increasing NaCl concentrations in the growth medium results in adverse effects on plant growth and survival. Salinity leads to severe impact on the physiology traits of wheat plants including total biomass; however, the sensitivity of salt impact varies greatly among wheat cultivars [32]. In the current study, a distinctive variation in salt tolerance was observed between two cultivars; Najran was more tolerant to saline cues than Qiadh. Under salt stress, a dramatic decline in mean fresh and dry weights as well as lengths of root and shoot tissues was measured in all cultivars, but the reduction was significantly lower in the NaCl-tolerant cultivar Najran than in the NaCl-sensitive cultivar Qiadh. The decline in these growth parameters might be a consequence of the increased salt concentration around the root area, which in turn causes water deficit, nutritional imbalance and osmotic stress in the plant, resulting in stomatal closure [33]. Moreover, prolonged exposure of plants to salinity leads to ion toxicity in the leaves and severely affects photosynthetic reactions, cell division and cell elongation, which results in a reduction in root and shoot lengths [34].

### 3.2. Yield, Salt Stress Impacted Differentially the Spike and Seed Number and Seed Weight and Germination

It is well known that salt stress negatively influences the expansion of plant leaves, causing a reduction in photosynthetic capacity, which in turn affects the quantity and quality of grain yield [35]. The results obtained in the present study confirm this, as the Qiadh cultivar showed the highest reduction in fresh and dry weights of the shoot, as well as in yield production, compared to the Najran cultivar. In contrast, salinity had a positive effect on spike and seed numbers in Najran, whereas seed weight had been negatively affected by NaCl. These findings are in line with those of [18], where a significant decline in yield outputs of all tested wheat cultivars except Sakha 94 and Sids 13 cultivars was found, suggesting that, while most wheat cultivars are sensitive to salinity, some cultivars are salt tolerant. Our results suggest that Najran wheat is among the salt-tolerant cultivars.

Interestingly, the Najran salt-tolerant cultivar seems to have invested carbon in producing more lighter seeds under salt stress while maintaining a relatively high rate of germination (around 90%). In the salt-sensitive Qiadh cultivar, the number of seeds produced under salinity was not the only factor to decline, but also the germination rate was reduced (around 70%). The fact that Najran wheat has chosen to maintain itself under salinity stress via seed production while maintaining a high germination rate testifies for its advanced adaptation to salt stress. It is crucial to find the genetic determinants of this trait. Several genes might be involved in the control of spike and seed numbers; the discovery of these genes is underway and some suspected transcription factors involved in the control of the developmental processes leading to spikes and seeds have already been identified, mainly in rice [36]. Considerable effort is being put into identifying the orthologs of these genes, as well as other genes potentially involved in the control of yield in wheat using mainly quantitative trait loci mapping.

### 3.3. Osmoregulation, Concomitant Production of Different Substances for Osmotic Adjustment Is Not Essentially Required for Salt Tolerance

Prolonged high salt concentrations impair plant growth due to the resulting hyperionic and hyperosmotic stresses. Plants respond to these stresses by implementing biochemical mechanisms to facilitate water uptake and therefore maintain cell turgor and plant growth [37]. Osmotic adjustment is one of the crucial biochemical strategies in plant acclimation to salt stress. Proline, soluble sugars, starch and organic acids are of the main organic osmotica which are synthesized within plants to assist survival under salt stress. It is demonstrated in this study that different *T. aestivum* cultivars might employ various mechanisms to alleviate the harmful effects of saline stress. For example, under salt stress, Qiadh had higher proline content in shoots than Najran; in contrast, Najran had higher proline content than Qiadh in roots. The results indicate that both cultivars responded to the high level of salt by increasing proline content mainly in shoots. Proline not only plays a great role in osmoregulation but also to protect plants from the damage caused by ROS and toxic ions. Proline has been found to participate in lowering osmotic potential [38], storing carbon and nitrogen [39], detoxifying ROS [40], protecting the enzyme activities of photosynthesis and production of antioxidants [41] and inducing adaptive responses by acting as a stress signal [42] under unfavorable conditions. Interestingly, transcript levels for P5CS1, a key gene in the production of proline, increased in roots and to low extent in shoots in Qiadh under salt stress, however they remained unchanged compared to the control in Najran plants. The non-correlation between levels of *P5CS1* transcripts and proline content might be explained by the fact that the gene might be differentially transiently expressed when the plant comes first under stress. The transiently produced transcripts lead to the production of enough P5CS1 enzyme to sustain the observed increase in proline. The transient increase in transcript levels might take place for different durations depending on the level of plant resistance to stress.

Soluble sugars increased in roots and shoots of both Najran and Qiadh cultivars under salt stress. Similar results were reported in [43], where a higher accumulation of soluble sugars was observed in the salt tolerant rice genotype, Pokkali, suggesting important role for soluble sugars in osmotic adjustment and accumulation of carbon reserves in plants under salt stress.

Similarly to soluble sugars the two cultivars exhibited an increase in starch accumulation. Najran displayed the highest starch levels in both roots and shoots of salt stressed plants. This accumulation may result from sustained photosynthetic activity required for maintaining plant biomass. In contrast [44] pointed out a decline in starch concentration in salt-treated leaves of *Oryza sativa* L. as a result of carbon limitation due to the poor photosynthetic activity under salt stress, the decline in this case might be a result of the suppression of starch biosynthesis. Increasing the level of soluble sugars and starch levels might be a critical trait in salt tolerant wheat. Transcript levels of key genes for the synthesis of soluble sugars (*SUS*) and starch (*STS*) increased in shoots in both cultivars and in roots of Qiadh indicating differential regulation at the gene levels between salt-tolerant and salt-sensitive wheat cultivars.

Organic acids are ubiquitous metabolites in plants which accumulate in response to salt stress to act as compatible solutes for osmoregulation and as ROS scavenger as well as plant protectors. The accumulation of total organic acids in the salt stressed wheat cultivars was obvious in the roots, and shoots of Najran and Qiadh cultivars. A previous study confirmed the increase in organic acids in root tissues, however the same study indicated a depletion in the leaves under saline treatment, suggesting that organic acids might be attributed to organ-specific functions [45]. In the case of salt stress, roots uptake excessive amounts of sodium cations which require anions to balance the charge. Thus, organic acids accumulated more in the roots to enhance the cation–anion balance. Furthermore, their high level in the roots assist plants to osmotically adjust under salinity conditions. The Najran cultivar accumulated higher levels of organic acids in both roots and shoots under salt stress compared to Qiadh, which is consistent with the higher salt-tolerance of this cultivar.

### 3.4. Antioxidant, Salt Stress Enhanced Phenolics Production as an Antioxidant Response

Salt stressed plants respond to oxidative stress resulting from the accumulations of ROS by operating an antioxidant-defense systems that prevent damage caused by ROS and detoxify ROS molecules. Phenolics are one of the nonenzymatic antioxidants produced to mainly protect plants against various stresses and act as ROS scavengers. The activity of the ROS-defense systems raise under extreme environmental stresses and is more pronounced in tolerant plants than sensitive ones [46,47]. This suggests that the defense systems perhaps work more effectively under unfavorable conditions. In the current study, prolonged saline stress caused significant accumulation of total phenolics in the two wheat cultivars which was more pronounced in Najran, confirming that wheat cultivars with different sensitivity to NaCl stress exhibit different levels of metabolites. Previous studies [48] have pointed out possible involvement of phenolics particularly phenylpropanoids in salt-tolerance in wheat. The expression of many of the genes involved in the productions of these substances was shown by transcriptomics to increase under salt stress in Najran wheat [49]. It would be of interest to extend transcriptomics analysis to the Qiadh cultivar and compare the results to those previously obtained in Najran. Interestingly, transcript levels for PAL, an important gene in the production of phenolics increased in shoots and roots of Qiadh but not Najran; even though level of phenolics under salt stress was higher in Najran. This might indicate differential regulation of the resistance mechanisms between wheat cultivars, in salt-resistant cultivars, resistance mechanisms like phenolics production are rapidly and transiently induced.

## 4. Materials and Methods

### 4.1. Plant Materials and Salt Stress Treatment

Seeds of two wheat (*Triticum aestivum*) genotypes, Najran, and Qiadh were obtained from the Ministry of Environment, Water and Agriculture, Saudi Arabia. Prior to sowing, seeds were stratified by incubation in the dark at 4 °C for 3 days to break seed dormancy and stimulate germination. Six cold-stratified seeds were sown in 2 L plastic pots filled with a mixture of John Innes soil compost No. 2, vermiculite 2–5 mm and grit sand in a volume ratio of 2:1:1, respectively. Pots were irrigated with either tap water for control plant-set, 100 mM NaCl solution for yield stage plant-set or 200 mM NaCl solution for seedling stage plant-set then sealed with cling film to maintain moisture. Pots were placed in a controlled growth cabinet under a 16 h light/8 h dark and 20 °C day/15 °C night and constant 70% humidity. After germination, three randomized seedlings from each pot were retained and watered 3 times a week. One-month old plants were harvested at midday to conduct growth and biochemical measurements. To assess the effect of salt stress on yield output a plants set was harvested after grain filling.

### 4.2. Growth and Yield Analysis

Root and shoot of thirty-day old Najran, and Qiadh wheat plants were harvested separately, and roots rinsed with tap water. Different growth parameters such as root length (RL), shoot length (SL), root fresh weight (RFW), shoot fresh weight (SFW), root dry weight (RDW) and shoot dry weight (SDW) were recorded. Roots and shoots of each cultivar were grouped into three replicates (each sample consisted of duplicate plants), then frozen in liquid nitrogen and stored at −80 °C after grinding them under liquid nitrogen to a fine powder to be used in biochemical analyses. Dry weight was determined after drying plant tissues in an oven at 80 °C for 2 days. To evaluate the extent to which the yield was affected by salinity, number of spikes, number of seeds per plant, seeds weight and seed germination rate were obtained.

### 4.3. Measurement of Proline Content

Total free proline content in the control and salt-stressed plants of the two wheat cultivars was measured using a modified colorimetric method described by [50]. Ground root and shoot samples from each treatment (100 mg each) were transferred to a 2 mL micro centrifuge tube, then homogenized in 1 mL of 3% (*w*/*v*) sulphosalicylic acid. The homogenate was clarified by centrifugation at 10,000× *g* for 3 min at room temperature. A volume of 500 µL of each supernatant was mixed with 500 µL of glacial acetic acid and 500 µL of acidic ninhydrin reagent in a 2 mL micro centrifuge tube. To make the nihydrin reagent, 2.5 g ninhydrin was dissolved in 100 mL of a solution made of 60 mL glacial acetic acid, 30 mL diH_2_O, and 10 mL 85% orthophosphoric acid. The reaction mixtures were incubated in a heat block at 98 °C for 1 h, then cooled at room temperature. After cooling, absorbance of the red colour developed in samples was read spectrophotometry at 546 nm. The concentration of proline in each sample was measured using a standard curve made using commercial pure L-proline and calculated on a dry weight basis (µg proline mg^−1^ DW).

### 4.4. Measurement of Soluble Sugars and Starch Level

Soluble and insoluble carbohydrates were quantified in salt-stressed and unstressed plants from all wheat cultivars using the phenol/sulphuric acid method based on a colorimetric assay. From ground root and shoot samples, 100 mg plant tissue was homogenized in 1 mL of 80% methanol in an Eppendorf tube and then heated at 80 °C for 40 min. The homogenate was centrifuged at 13,000 rpm for 10 min at room temperature, then supernatant was transferred to a new tube to be used in soluble sugar assay, and the remaining plant tissue was kept for measuring starch level. To extract starch, the remaining tissue was washed several times with acetate buffer to remove any traces of glucose. After that, 1.2 mL acetate buffer and 0.2 mL enzyme cocktail were added to digest starch molecules into a glucose equivalent, and incubated overnight at 45 °C. For enzyme cocktail, 26 mg (300 units) amyloglucosidase and 9 mg (25 units) amylase (Sigma-Aldrich, Dorset, UK) were mixed in 20 mL acetate buffer. After incubation, the homogenate was centrifuged at 13,000 rpm for 10 min at room temperature. Exactly 0.5 mL of each supernatant prepared for either soluble sugar or starch assays was transferred to a glass tube, then 0.5 mL diH_2_O, 0.5 mL 5% phenol and 2.5 mL sulphuric acid were added, respectively, and left to cool for 15 min at room temperature. The absorbance of reaction mixtures was read using a spectrophotometer at 483 nm and then plotted against a standard curve created using commercial glucose with different known concentrations.

### 4.5. Measurement of Total Organic Acids

The content of organic acids in the root and shoot tissues from the control and salt-treated plants was assessed using a basic titration method. Plant tissues (100 mg) were homogenized in 1 mL 80% methanol and incubated at 80 °C for 40 min. The extracts were centrifuged at 13,000 rpm for 10 min and the supernatants were collected. A 20 µL aliquot of plant extract was transferred to a small vial; 970 µL distilled water and 10 µL phenolphthalein (10 mg·mL^−1^) as a pH indicator were added to the exact. The total acidity of the mixture was neutralized with 0.1 N sodium hydroxide, added from a titration burette, until a pink color was obtained. The volume of sodium hydroxide used was obtained by reading the burette and the titration data were calculated and expressed on a dry weight basis (µmol·mg^−1^ DW).

### 4.6. Measurement of Phenolics Content

Total phenolics in root and shoot plant materials of the two different wheat cultivars were estimated using the Folin–Ciocalteu (F-C) reagent. To 20 µL of plant extracts, prepared in a previous experiment and stored at −20 °C, 200 µL of 10% F-C reagent and 800 µL of 0.7 M Na_2_CO_3_ were added and mixed thoroughly in a 2 mL tube. The mixture tubes were incubated at room temperature for 120 min. After incubation, tube content was transferred to cuvettes and absorbance readings were taken using a spectrophotometer at 265 nm. The levels of phenolic compounds were determined from a standard curve plotted using gallic acid.

### 4.7. RNA Extraction and qRT-PCR

Total RNA was isolated from shoot and root ground tissues using the tri-reagent method, as previously described [51]. About 200 mg of ground leaf tissue were extracted in 1 mL of tri-reagent; the obtained total RNA was DNaseI-treated to remove contaminating DNA, then quantified and qualified using a nanodrop spectrophotometer. cDNA was synthesized from purified RNA samples using a Revert Aid reverse transcription kit (ThermoFisher, Paisley, UK) according to the manufacturer’s procedure using Oligo dT primer. The obtained cDNAs were quantified using a nanodrop spectrophotometer and equal amounts per sample were used to amplify by real-time PCR both the target genes as well as the reference genes using gene-specific primers (Table 1). The SensiFAST™ SYBR Hi-ROX Kit (Meridian Biosciences, London, UK) was used to perform qPCR in 20 µL reactions containing 1× Sensi Fast master mix, 100 nM forward and reverse primers each and 100 ng cDNA. The PCRs were run in a CFX96 C1000 real PCR machine (Bio-Rad, Watford, UK) using the following program: 98 °C/3 min; [98 °C/10 s, 58 °C/15 s, 72 °C/30 s] × 39 times; 72 °C/5 min. Normalization of gene expression level was performed using the Gapdh gene as a housekeeping (internal control) gene from wheat. Primer efficiency was determined using serial dilutions of the cDNA for each gene; it ranged from 97% to 98%. qRT-PCRs were run twice on two biological replicates and the mean Ct was used to calculate relative gene transcript levels of the target genes using the standard 2^−(∆∆Ct)^ method, where the Ct of the unstressed control samples was subtracted from the Ct of the salt-treated samples for the target genes and normalized to delta Ct of the reference gene (*GAPDH*) [52]).

### 4.8. Statistical Analysis

To confirm the significance of differences between salt-treated plants and control plants, a paired T test was conducted and standard errors were calculated based on at least 6 replicates using the SPSS statistical software (Version 29, Chicago, IL, USA).

## 5. Conclusions

Salt stress caused differential reduction in physiological activities and grain yield depending on wheat cultivar. Moreover, shoot and root tissues from the two different wheat cultivars with different sensitivity to NaCl stress exhibited different metabolic alterations and antioxidant responses. These salinity effects were less pronounced in the Najran cultivar, potentially due to its high osmotic and antioxidant responses; therefore, it was characterized as the most tolerant cultivar to salt stress. Interestingly, transcript levels for key genes involved in osmoregulation (P5CS1) and antioxidant (PAL) increased in the roots and shoots of the sensitive cultivar, while showing little change in the tolerant cultivar under stress. Transcript for genes involved in sugar metabolism (SUS and STS) increased in the shoots and roots of the sensitive cultivar while they decreased in roots and increased in shoots of the resistant cultivar. This difference in gene response to salt stress between cultivars suggests important differences in gene regulation, with the resistant wheat cultivars exhibiting a transient expression of the resistance genes. Given its relatively high salt tolerance, the Najran cultivar might be used in breeding program for creating novel wheat cultivars with improved salt tolerance. Crosses between Najran wheat and other cultivars having useful traits (e.g., yield, grain quality, resistance to disease, etc.) would lead to the creation of superior cultivars with higher salt tolerance.

## Figures and Tables

**Figure 1 ijms-26-03742-f001:**
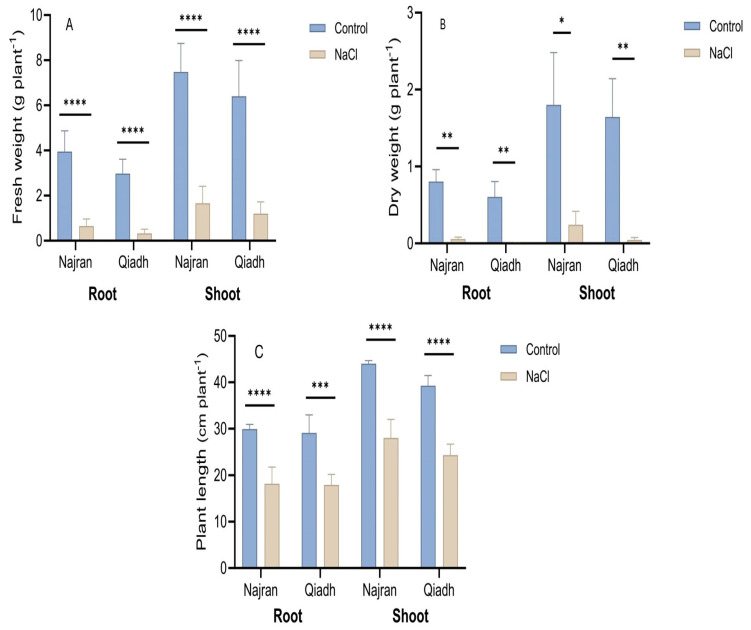
Effect of salt stress on (**A**) root and shoot fresh weight root and (**B**) shoot dry weight, and (**C**) root and shoot lengths of two wheat (*T. aestivum*) cultivars, Najran, and Qiadh (n = 6 ± S.E). Salt-treated plants were watered with 200 mM NaCl, whereas control plants were watered with 0 mM NaCl (tap water). Asterisks refer to significant differences at confidence levels of * *p* < 0.05, ** *p* < 0.01, *** *p* < 0.001 and **** *p* < 0.0001.

**Figure 2 ijms-26-03742-f002:**
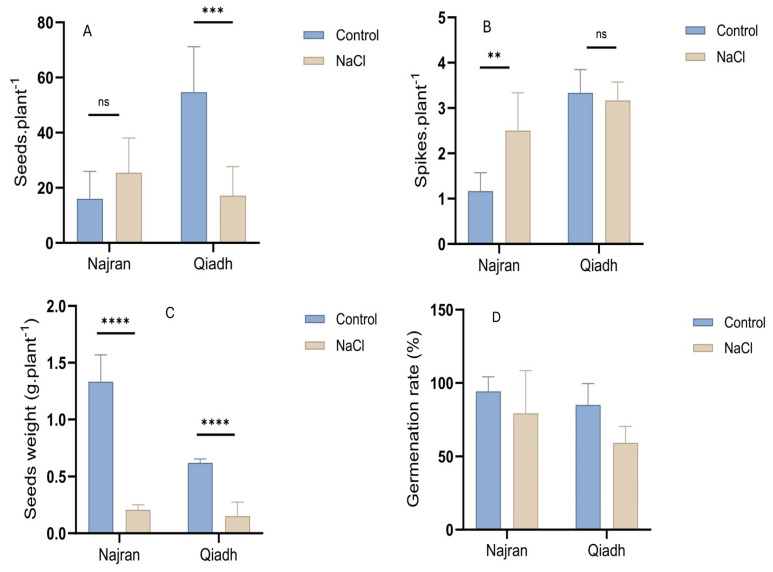
Effect of salt stress on (**A**) spike number, (**B**) seed number, (**C**) seed weight and (**D**) germination rate of two wheat (*T. aestivum*) cultivars, Najran and Qiadh. Salt-treated plants were watered with 200 mM NaCl, whereas control plants were watered with 0 mM NaCl (tap water). Asterisks refer to significant differences at confidence levels of ** *p* < 0.01, *** *p* < 0.001 and **** *p* < 0.0001; ns: no significance.

**Figure 3 ijms-26-03742-f003:**
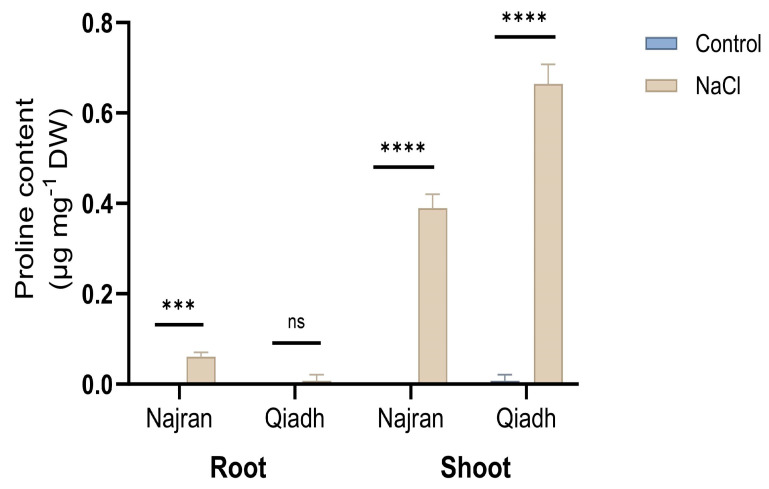
Effect of salt stress on proline content in roots and shoots of two wheat (*T. aestivum*) cultivars, Najran, and Qiadh (n = 3 ± S.E). Salt-treated plants were watered with 200 mM NaCl, whereas control plants were watered with 0 mM NaCl (tap water). Asterisks refer to significant differences at confidence levels of *** *p* < 0.001 and **** *p* < 0.0001; ns: no significance.

**Figure 4 ijms-26-03742-f004:**
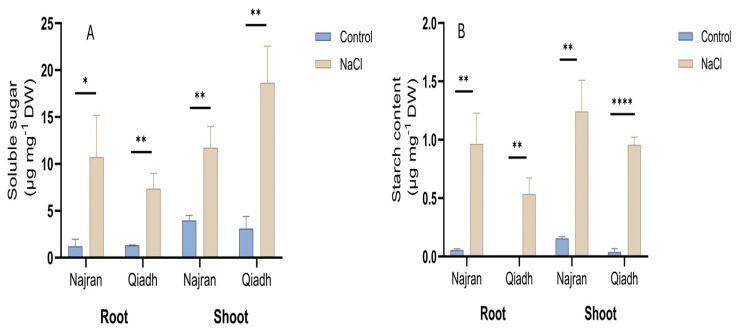
Effect of salt stress on levels of soluble sugars (**A**) and starch (**B**) in the roots and shoots of two wheat (*T. aestivum*) cultivars, Najran and Qiadh (n = 3 ± S.E). Salt-treated plants were watered with 200 mM NaCl, whereas control plants were watered with 0 mM NaCl (tap water). Asterisks refer to significant differences at confidence levels of * *p* < 0.05, ** *p* < 0.01 and **** *p* < 0.0001.

**Figure 5 ijms-26-03742-f005:**
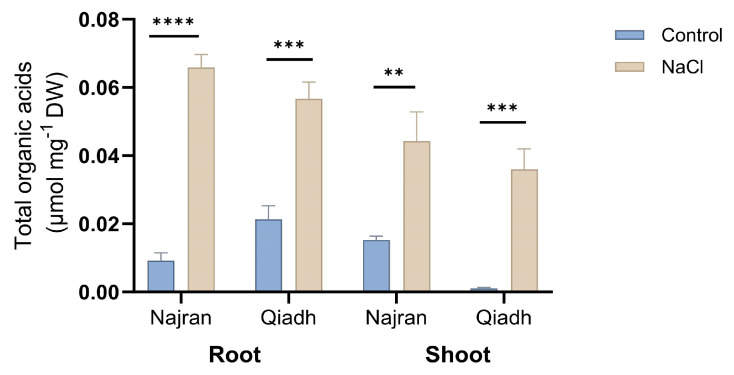
Effect of sat stress on the content of total organic acids in the roots and shoots of two wheat (*T. aestivum*) cultivars, Najran and Qiadh (n = 3 ± S.E). Salt-treated plants were watered with 200 mM NaCl, whereas control plants were watered with 0 mM NaCl (tap water). Asterisks refer to significant differences at confidence levels of ** *p* < 0.01, *** *p* < 0.001 and **** *p* < 0.0001.

**Figure 6 ijms-26-03742-f006:**
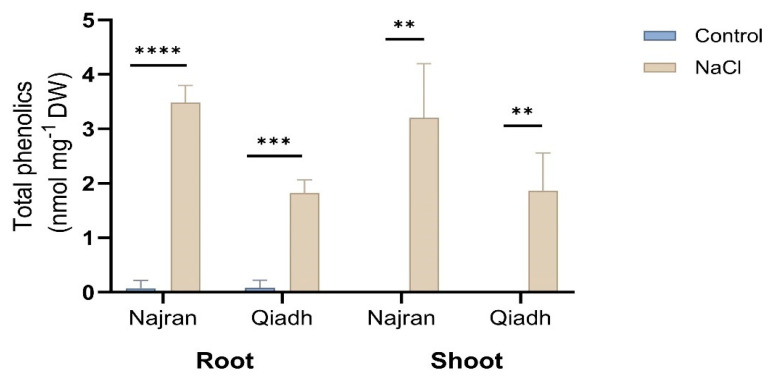
Effect of salt stress on total phenolics content in roots and shoots of two wheat (*T. aestivum*) cultivars, Najran and Qiadh (n = 3 ± S.E). Salt-treated plants were watered with 200 mM NaCl, whereas control plants were watered with 0 mM NaCl (tap water). Asterisks refer to significant differences at confidence levels of ** *p* < 0.01, *** *p* < 0.001 and *** *p* < 0.0001.

**Figure 7 ijms-26-03742-f007:**
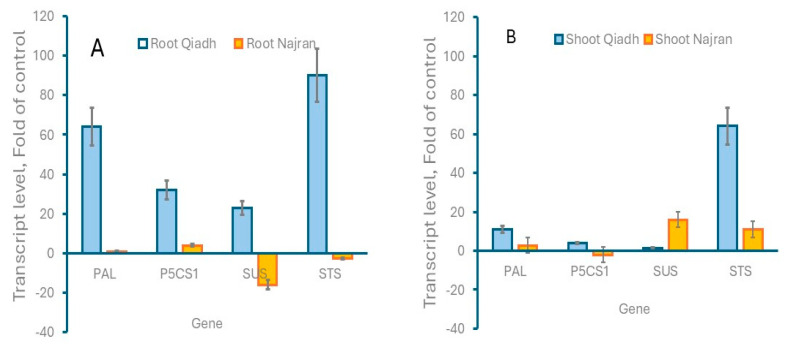
Effect of salt stress on transcript levels for phenylalanine ammonia lyase (PAL), delta1-1-pyrroline-5-carboxylate synthase (P5CS1), sucrose synthase (SUS) and starch synthase (STS) in roots (**A**) and shoots (**B**) of two wheat (*T. aestivum*) cultivars, Najran and Qiadh (n = 4 ± S.E). Salt-treated plants were watered with 200 mM NaCl, whereas control plants were watered with 0 mM NaCl (tap water).

**Table 1 ijms-26-03742-t001:** Primer sets used for qRT-PCR semi-quantification of transcript levels for delta1-1-pyrroline-5-carboxylate synthase (P5CS1), phenylalanine ammonia lyase (PAL), sucrose synthase (SUS) and starch synthase (STS) using Glyceraldehyde-3-phosphate dehydrogenase (GAPDH) as control in roots and shoots of Najran and Qiadh wheat.

Gene	Primers	Melting Temperature (°C)	Efficiency (%)	Amplicon Size (bp)	GenBank, Number
*PAL*	Forward: 5′TAGTGTTGGGTAGCCAGTAGA3′	62	97.5	136	*X99725.1*
	Reverse: 5′AGTTGGTTCTCGGACTATTGC3′	62			
*P5CS1*	Forward: 5′CGCGAAACTGTCGAGTCATTAT3′	63	97.2	130	*XM_048685323*
	Reverse: 5′ACCTGCCAAACTGTCATTATCC3′	63			
*SUS*	Forward:5′GAAGTACGTGAGCAACCTAGAG3′	62	98	105	*XM_048673430.1*
	Reverse: 5′TCAACCGCCAATGGAACT 3′	62			
*STS*	Forward: 5′ACGTGCTTCTGGAACTGG3′	62	97.4	120	*AY050174.1*
	Reverse: 5′CCAGAAGCTCCTCTTCTTTCTC3′	63			
*GAPDH*	Forward: 5′AACGACCCCTTCATCACCAC3′	65	98	150	*EF592180*
	Reverse: 5′GTTCCTGCAGCCAAACACAG3′	65			

## Data Availability

The datasets used and/or analyzed during the current study are available from the corresponding author on reasonable request.

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
