# Peer review of "Comparative Analysis of Physiological and Biochemical Responses to Salt Stress Reveals Important Mechanisms of Salt Tolerance in Wheat"

_ijms, 2025, doi:10.3390/ijms26083742_

Round 1
Reviewer 1 Report
Comments and Suggestions for Authors
The research done is justified. The introduction is well written. The methodology is also well described. Only some methods are very old (70 years old). In the European Union, there are appropriate standards for the methods and cut-off numbers used. For some methods, there is a lack of information regarding the position in references. This needs to be supplemented. Volume is given in cm3 not ml. Figures are made correctly. 15 specific notes are highlighted in the text.

Author Response
Dear Reviewer
Many thanks for taking the time to review our manuscript, we appreciate your constructive comments, we have made changes to the manuscript that we hope meet your comments, would you please check the uploaded document to see the changes listed and explained point-by-point.
Regards
Dr. Taybi

Reviewer 2 Report
Comments and Suggestions for Authors
The authors of the manuscript "Comparative Analysis of Physiological and Biochemical Responses to Salt-Stress Reveals Important Mechanisms of Salt-Tolerance in Wheat" present an interesting study on the impact of salt stress on two Triticum aestivum cultivars that respond differently to excess salt. Growth parameters, biochemical responses, antioxidants, and gene regulation were analyzed in depth. The authors identify the most tolerant cultivar (Najran) as a potential source of genes for genetic improvement programs in T. aestivum. The manuscript is generally clearly written, and the results are interesting, but there are minor flaws that need improvement.
Points that need to be addressed as “Minor Issues”:
Keywords section: authors must add them.
Line 94: there are actually two cultivars, not three. Authors must correct the number of cultivars that were written.
Section 2: authors should standardize the volume units by adopting capital letter for "liter," i.e., mL or µL.
Line 104: are it 2 or 3 cultivars? Authors must clarify.
Line 214: what are FWt and DWt? If there are typos, please correct them. If they have other meanings, please add brief descriptions.
Line 219: the specie name can be written in abbreviated form because it is already mentioned. The same correction will be applied to all of the following figures.
Line 247: authors could write "control conditions" instead of "unstressed conditions."
Section 4.1 title: Why three? It should be changed.
Author Response
Dear Reviewer
Thank you very much for taking the time to review out manuscript, we have considered your constructive comments and made change to meet them. Would you please check the uploaded document to see the made changes listed point-by-point.
Regards
Dr. Taybi

Round 2
Reviewer 1 Report
Comments and Suggestions for Authors
The authors have corrected almost all errors. References are still incorrectly edited. The second remark is degrees Celsius are written without spaces.

Author Response
The authors have corrected almost all errors. References are still incorrectly edited. The second remark is degrees Celsius are written without spaces.
Dear Reviewers
We have edited the references to the IJMS suggested format and added a space between the temperature numbers and Celsius. all changes are shown by track changes in the second revision.
Thank you again for taking the time to review our manuscript and provide very constructive recommendations.
Regards
Dr. Taybi